# Decomposed Knowledge Distillation for Class-Incremental Semantic Segmentation

Donghyeon Baek[1]       Youngmin Oh[1]       Sanghoon Lee[1]
Junghyup Lee[1]       Bumsub Ham[1,2*]

[1]Yonsei University       [2]Korea Institute of Science and Technology (KIST)

https://cvlab.yonsei.ac.kr/projects/DKD/

## Abstract

Class-incremental semantic segmentation (CISS) labels each pixel of an image with a corresponding object/stuff class continually. To this end, it is crucial to learn novel classes incrementally without forgetting previously learned knowledge. Current CISS methods typically use a knowledge distillation (KD) technique for preserving classifier logits, or freeze a feature extractor, to avoid the forgetting problem. The strong constraints, however, prevent learning discriminative features for novel classes. We introduce a CISS framework that alleviates the forgetting problem and facilitates learning novel classes effectively. We have found that a logit can be decomposed into two terms. They quantify how likely an input belongs to a particular class or not, providing a clue for a reasoning process of a model. The KD technique, in this context, preserves the sum of two terms (*i.e.*, a class logit), suggesting that each could be changed and thus the KD does not imitate the reasoning process. To impose constraints on each term explicitly, we propose a new decomposed knowledge distillation (DKD) technique, improving the rigidity of a model and addressing the forgetting problem more effectively. We also introduce a novel initialization method to train new classifiers for novel classes. In CISS, the number of negative training samples for novel classes is not sufficient to discriminate old classes. To mitigate this, we propose to transfer knowledge of negatives to the classifiers successively using an auxiliary classifier, boosting the performance significantly. Experimental results on standard CISS benchmarks demonstrate the effectiveness of our framework.

## 1 Introduction

A general way of learning knowledge for neural networks is to tune network weights with examples for all object/scene classes at hand. After finishing the learning process, the weights are normally fixed for inference, suggesting that the current learning paradigm is not flexible enough to handle novel classes unseen at training time. Fine-tuning the weights with additional examples for novel classes addresses the problem in part, but this causes *catastrophic forgetting* [19]. Namely, neural networks rather forget the previously learned knowledge in order to learn new information. Class-incremental learning (CIL) targets to learn novel object/scene classes continually using training samples for those classes only, while minimizing the forgetting problem. A key to CIL is to design a learning method that balances between *rigidity* and *plasticity* of a model [20]. On the one hand, network weights should not be altered abruptly in learning new information from novel classes, in order to preserve the discriminative ability for old ones (*i.e.*, rigidity), avoiding catastrophic forgetting. On the other

---

*Corresponding author

36th Conference on Neural Information Processing Systems (NeurIPS 2022).

hand, a strong rigidity rather distracts learning knowledge from novel classes. The network weights should thus be tuned accordingly (*i.e.*, plasticity).

Class-incremental semantic segmentation (CISS) adopts a CIL paradigm for the task of semantic segmentation. CISS methods [5, 9, 21, 29] typically exploit a softmax cross-entropy (CE) term along with knowledge distillation (KD) [14]. Although the CE term helps to learn novel classes, applying the softmax function to all classes, including both old and novel ones, lowers class probabilities of old ones. This in turn prevents the model from preserving knowledge learned from old classes, resulting in catastrophic forgetting [1, 23]. The KD technique prevents changing network weights drastically, alleviating the forgetting problem. Recently, SSUL [6] proposes to use multiple binary cross-entropy (BCE) terms for individual novel classes separately. This approach handles the forgetting problem caused by the softmax function, but it is limited in the following: (1) A feature extractor is frozen in order to enforce the rigidity of the model. This strong constraint for preserving knowledge for old classes makes it hard to learn discriminative features for novel classes. (2) An initialization technique for classifiers requires an off-the-shelf saliency detector [15], which is computationally demanding.

In this paper, we present a simple yet effective CISS framework that overcomes the aforementioned problems. To achieve better plasticity and rigidity for a CISS model, we propose to train a feature extractor, and introduce a decomposed knowledge distillation (DKD) technique. KD encourages a model to predict logits similar to the ones obtained from an old model. We have found that logits can be represented as the sum of *positive and negative reasoning scores* that quantify how likely and unlikely an input belongs to a particular class, respectively. In this context, KD focuses on preserving the relative difference between positive and negative reasoning scores only, without considering the change of each score. The DKD technique imposes explicit constraints on each reasoning score, instead of class logits themselves, which is beneficial to improving the rigidity of a CISS model together with KD effectively. We also propose an initialization technique to train classifiers for novel classes effectively. Note that training samples for novel classes are available only for each incremental step, suggesting that classifiers for the novel classes are trained with a small number of negative samples. To address this, we propose to train an auxiliary classifier in a current step, and use it to initialize classifiers for novel classes in a next step. To this end, we consider training samples of a current step as potential negatives for novel classes in a next step. We then train the auxiliary classifier, such that all pixels in current training samples as negative ones, transferring prior knowledge of negative samples to the next step for the classifiers of novel classes. Our initialization technique also does not require any pre-trained models, *e.g.*, for saliency detection [6], in order to differentiate possibly negative samples. We demonstrate the effectiveness of our framework with extensive ablation studies on standard CISS benchmarks [11, 30]. We summarize main contributions of our work as follows:

- We introduce a simple yet effective CISS framework that exploits a novel DKD and multiple BCE terms, achieving a good trade-off between rigidity and plasticity.
- We present a novel initialization technique that encodes prior knowledge for negatives to train classifiers for novel classes effectively.
- We achieve a new state of the art on standard benchmarks for CISS [11, 30], and demonstrate the effectiveness of our approach through extensive experiments and ablation studies.

## 2 Related work

### 2.1 Class incremental image classification

Many CIL methods [2, 4, 8, 10, 16, 27] have been proposed for image classification, attempting to preserve the discriminative ability for old classes. Since training samples of novel classes are available only in incremental steps, they typically adopt a KD [14] technique to retain classifier logits for old classes. For example, the works of [8, 10] additionally apply the technique to intermediate feature maps from a feature extractor. LwM [8] proposes an attention-based distillation loss to preserve visual information of old classes. PODNet [10] introduces a spatial-based distillation technique that encourages pooled feature maps between current and previous steps to be similar to each other. CIL methods [2, 4, 16, 27] exploiting an external memory have recently been introduced. They store a subset of training samples for old classes for re-training, which is effective to alleviate catastrophic forgetting. Due to the data imbalance between old and novel classes, they generally use training

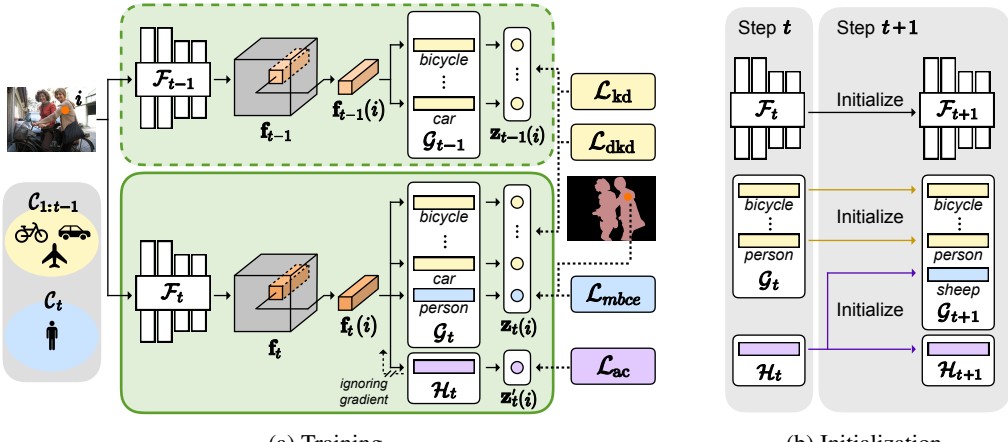

| (a) Training. | (b) Initialization. |
|---|---|

Figure 1: Overview of our framework. (a) Our framework consists of a feature extractor $\mathcal{F}_t$, classifiers $\mathcal{G}_t$, and an auxiliary classifier $\mathcal{H}_t$ at each step $t$. Given an input image, we extract a feature map $\mathbf{f}_t$, and obtain class logits $\mathbf{z}_t$ from corresponding classifiers $\mathcal{G}_t$. We train our model with four terms: mBCE ($\mathcal{L}_{\mathrm{mbce}}$), KD ($\mathcal{L}_{\mathrm{kd}}$), DKD ($\mathcal{L}_{\mathrm{dkd}}$), and AC ($\mathcal{L}_{\mathrm{ac}}$) losses. Note that the feature extractor does not receive any gradients from the AC term for the auxiliary classifier. (b) In the next step $t+1$, we initialize classifiers for novel classes with the previous auxiliary classifier $\mathcal{H}_t$. The feature extractor $\mathcal{F}_{t+1}$, a new auxiliary classifier $\mathcal{H}_{t+1}$, and other classifiers are simply initialized with the counterparts from the step $t$. Best viewed in color.

tricks, such as re-training with a balanced subset of training samples [4, 27] or re-balancing classifier weights [2, 16], which is however computationally expensive and requires additional memory. Note that all the aforementioned methods [2, 4, 8, 10, 16, 27] exploit a softmax CE term to learn novel classes. Similar to ours, the seminal work of [25] uses BCE losses along with KD for CIL. Differently, we use a novel DKD loss together with a novel initialization technique specialized for CISS.

## 2.2 Class incremental semantic segmentation

Similar to class-incremental classification, CISS methods [5, 9] generally adopt the KD technique to alleviate catastrophic forgetting. For example, MiB [5] applies the technique to pixel-wise classification scores. PLOP [9] employs KD to intermediate feature maps, and introduces a local POD loss, specially designed for CISS, by extending the vanilla version in [10]. In addition, SDR [22] recently proposes to leverage contrastive learning, complementary to existing KD techniques, to separate feature clusters, making it easier to learn novel classes. These methods also exploit a softmax CE term to learn novel classes. Different from image classification, supervisory signals for CISS are given in pixel-level labels only for regions corresponding to novel classes. The regions for background and old classes are thus unlabeled, suggesting that we have limited information to train classifiers for CISS using a softmax CE loss. To tackle this issue, several works [9, 22] propose to generate pseudo labels from an old model to provide auxiliary supervisory signals for the unlabeled regions. Following the work of [25], exploiting multiple BCE losses for CISS is recently introduced to train classifiers for novel classes individually [6]. This approach enables training a model without auxiliary supervisory signals for unlabeled regions. It however freezes a feature extractor to alleviate catastrophic forgetting, constraining the plasticity of classifiers for novel classes excessively. On the contrary, our method trains both a feature extractor and classifiers, together with novel DKD and initialization techniques, achieving a better trade-off in terms of plasticity and rigidity.

## 3 Approach

We train a CISS model continually with training samples $D_t$ at each training step $t$, where $t \in \{1, ..., T\}$ and $T$ is a total number of steps. The training dataset $D_t$ contains pairs of an image and a corresponding ground-truth mask $\mathbf{y}$. We denote by $\mathcal{C}_t$ and $\mathcal{C}_{1:t-1}$ sets of novel and old object/stuff classes at the step $t$, respectively. Note that the sets are disjoint, *i.e.*, $\mathcal{C}_{1:t-1} \cap \mathcal{C}_t = \emptyset$. Note also that regions for background and old classes are labeled as *unknown* in a current step $t$. Namely, a ground-truth label at position $i$, $\mathbf{y}(i)$, is either one of the classes in $\mathcal{C}_t$ or the unknown class $c_u$, *i.e.*, $\mathbf{y}(i) \in \mathcal{C}_t \cup \{c_u\}$, at the step $t$.

### 3.1 Overview

We show in Fig. 1 an overview of our framework. For each step $t$, we train a CISS model together with an auxiliary classifier $\mathcal{H}_t$. Our framework mainly consists of a feature extractor $\mathcal{F}_t$ and a set of classifiers $\mathcal{G}_t$ predicting pixel-level semantic labels for previous and novel classes at the step $t$ (Fig. 1(a)). In a next step $t+1$, we initialize a feature extractor $\mathcal{F}_{t+1}$ and classifiers for old classes $\mathcal{C}_{1:t}$ with the previous ones from the step $t$ to preserve knowledge for old classes. Classifiers for novel classes at the step $t+1$ and a new auxiliary classifier $\mathcal{H}_{t+1}$ are initialized with the previous one $\mathcal{H}_t$ (Fig. 1(b)).

We exploit four loss terms for training (Fig. 1(a)): Multiple binary cross-entropy (mBCE), knowledge distillation (KD), decomposed knowledge distillation (DKD), and auxiliary classifier (AC) losses. The first three terms are used to train a CISS model at every step, while the last one is for the auxiliary classifier $\mathcal{H}_t$. Specifically, the mBCE term encourages the model to learn knowledge from novel classes. The KD and DKD terms, on the other hand, help to preserve the discriminative ability for old classes. The AC term enables transferring knowledge of negatives to the next step for classifiers of novel classes.

### 3.2 Training

We train our framework using an objective as follows:

$$\mathcal{L} = \mathcal{L}_{\text{mbce}} + \alpha \mathcal{L}_{\text{kd}} + \beta \mathcal{L}_{\text{dkd}} + \mathcal{L}_{\text{ac}}, \tag{1}$$

where $\mathcal{L}_{\text{mbce}}$, $\mathcal{L}_{\text{kd}}$, $\mathcal{L}_{\text{dkd}}$, and $\mathcal{L}_{\text{ac}}$ are mBCE, KD, DKD, and AC terms, respectively, balanced by the hyperparameters of $\alpha$ and $\beta$. In the following, we describe each term in detail.

**mBCE loss.** Given an input image, we first obtain a feature map $\mathbf{f}_t$ at a step $t$. We then compute class logits, $\mathbf{z}_t \in \mathbb{R}^{HW \times |\mathcal{C}_{1:t}|}$, with classifiers $\mathcal{G}_t$, where $H$ and $W$ are the height and width of the input image, respectively, and $|\cdot|$ is the cardinality of a given set. Concretely, the class logit is obtained by computing the dot product between features and weights for corresponding classifiers, followed by adding a bias:

$$\mathbf{z}_t(i,c) = \mathbf{f}_t(i)^\top \mathbf{w}_t(c) + \mathbf{b}_t(c), \tag{2}$$

where we denote by $\mathbf{w}_t(c)$ and $\mathbf{b}_t(c)$ weights and a bias of a classifier for a class $c$, respectively, and $\mathbf{f}_t(i)$ is a feature at position $i$. To learn novel classes of $\mathcal{C}_t$, current CISS methods typically exploit a CE loss computing softmax probabilities w.r.t all classes, $\mathcal{C}_t$ and $\mathcal{C}_{1:t-1}$, at a step $t$. This could lower the softmax probabilities for the old classes $\mathcal{C}_{1:t-1}$, causing catastrophic forgetting [1, 23]. We instead exploit a mBCE loss, and apply it to train classifiers for the novel classes $\mathcal{C}_t$ only as follows:

$$\mathcal{L}_{\text{mbce}} = -\frac{1}{HW} \sum_{i=1}^{HW} \sum_{c \in \mathcal{C}_t} \gamma \mathbb{1}[\mathbf{y}(i) = c] \log \mathbf{p}_t(i,c) + \mathbb{1}[\mathbf{y}(i) \neq c] \log\big(1 - \mathbf{p}_t(i,c)\big), \tag{3}$$

where

$$\mathbf{p}_t(i,c) = \frac{1}{1 + e^{-\mathbf{z}_t(i,c)}}, \tag{4}$$

and $\mathbb{1}[\cdot]$ is an indicator function that outputs 1 if the argument is true, and 0 otherwise. Following [3, 28], we use a weighting strategy with a balance parameter of $\gamma$ to handle the imbalance between two terms in (3).

**KD loss.** We adopt a KD term to prevent our model from changing abruptly, mitigating the catastrophic forgetting problem, defined as follows:

$$\mathcal{L}_{\text{kd}} = -\frac{1}{HW} \sum_{i=1}^{HW} \sum_{c \in \mathcal{C}_{1:t-1}} \mathbf{p}_{t-1}(i,c) \log \mathbf{p}_t(i,c) + \big(1 - \mathbf{p}_{t-1}(i,c)\big) \log\big(1 - \mathbf{p}_t(i,c)\big), \tag{5}$$

where $\mathbf{p}_{t-1}$ is similarly computed as in (4) with $\mathbf{z}_{t-1}$, *i.e.*, class logits predicted by a previous model at the step $t-1$. This term encourages our model to provide class logits similar to the ones obtained from the previous model, namely, $\mathbf{z}_t(i,c) \approx \mathbf{z}_{t-1}(i,c)$, for old classes, $c \in \mathcal{C}_{1:t-1}$, at the step $t$.

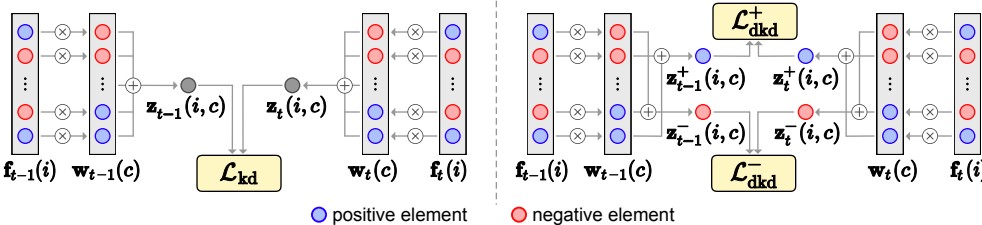

● positive element    ● negative element

Figure 2: Comparison of KD and DKD. KD uses a logit $\mathbf{z}_t$, while DKD exploits positive and negative reasoning scores, $\mathbf{z}_t^+$ and $\mathbf{z}_t^-$. The DKD term encourages our model to output the reasoning scores of $\mathbf{z}_t^+$ and $\mathbf{z}_t^-$, similar to the ones of $\mathbf{z}_{t-1}^+$ and $\mathbf{z}_{t-1}^-$, respectively, obtained from a previous model.

**DKD loss.** Based on that the dot product is the sum of element-wise multiplication between vectors, we decompose the class logit in (2) as follows:

$$\mathbf{z}_t(i,c) = \mathbf{z}_t^+(i,c) + \mathbf{z}_t^-(i,c), \tag{6}$$

where $\mathbf{z}_t^+(i,c)$ is the sum of positive elements chosen from the result of element-wise multiplication between $\mathbf{f}_t(i)$ and $\mathbf{w}_t(c)$, and $\mathbf{z}_t^-(i,c)$ is similarly defined using negative elements (Fig. 2 right). We omit the bias of the classifier for ease of notation. We call $\mathbf{z}_t^+(i,c)$ and $\mathbf{z}_t^-(i,c)$ as *positive* and *negative reasoning scores*, respectively, which quantify how likely and unlikely an input belongs to the class $c$. In this context, the KD technique retains the relative difference between positive and negative reasoning scores only, suggesting that each reasoning score itself is not preserved (Fig. 2 left). For example, if one of the reasoning scores increases, the other one would decrease accordingly in order to maintain the sum of reasoning scores, *i.e.*, the class logit. To address this problem, we propose a decomposed knowledge distillation (DKD) loss as follows:

$$\mathcal{L}_{\text{dkd}} = \mathcal{L}_{\text{dkd}}^+ + \mathcal{L}_{\text{dkd}}^-, \tag{7}$$

where

$$\mathcal{L}_{\text{dkd}}^+ = -\frac{1}{HW} \sum_{i=1}^{HW} \sum_{c \in \mathcal{C}_{1:t-1}} \mathbf{p}_{t-1}^+(i,c) \log \mathbf{p}_t^+(i,c) + \big(1 - \mathbf{p}_{t-1}^+(i,c)\big) \log\big(1 - \mathbf{p}_t^+(i,c)\big), \tag{8}$$

and

$$\mathbf{p}_t^+(i,c) = \frac{1}{1 + e^{-\mathbf{z}_t^+(i,c)}}. \tag{9}$$

$\mathcal{L}_{\text{dkd}}^-$ is similarly defined using the negative reasoning score $\mathbf{z}_t^-$. Note that $\mathcal{L}_{\text{dkd}}^+$ and $\mathcal{L}_{\text{dkd}}^-$ are analogous to the KD term in (5), but they compute losses with the reasoning scores, $\mathbf{z}_t^+$ and $\mathbf{z}_t^-$, separately, instead of exploiting a class logit $\mathbf{z}_t$ directly. That is, the DKD term encourages our model to provide positive and negative reasoning scores similar to the ones obtained from a previous model, *i.e.*, $\mathbf{z}_t^+(i,c) \approx \mathbf{z}_{t-1}^+(i,c)$ and $\mathbf{z}_t^-(i,c) \approx \mathbf{z}_{t-1}^-(i,c)$, for the old classes $c \in \mathcal{C}_{1:t-1}$. This explicit constraint on each reasoning score improves the rigidity of our model, enabling it to preserve the discriminative ability for old classes effectively.

**AC loss.** The dataset $D_t$ at a step $t$ provides training images mainly depicting one of novel classes $\mathcal{C}_t$. This suggests that the number of negative samples for the novel classes $\mathcal{C}_t$, *e.g.*, images containing objects for old classes $\mathcal{C}_{1:t-1}$ as well, would not be sufficient in the dataset $D_t$. We conjecture that training samples in the step $t$ can serve as good negative examples for novel classes $\mathcal{C}_{t+1}$ in the next step $t+1$, as $\mathcal{C}_{1:t} \cap \mathcal{C}_{t+1} = \emptyset$. Based on this, we propose to exploit an auxiliary classifier $\mathcal{H}_t$ to encode knowledge of the negatives for the novel classes $\mathcal{C}_{t+1}$ in advance of the step $t+1$. To this end, we train the classifier $\mathcal{H}_t$ with the AC term as follows:

$$\mathcal{L}_{\text{ac}}(\mathbf{p}_t') = -\frac{1}{HW} \sum_{i=1}^{HW} \log\big(1 - \mathbf{p}_t'(i)\big), \tag{10}$$

where $\mathbf{p}_t'(i) = 1/(1 + e^{-\mathbf{z}_t'(i)})$, and $\mathbf{z}_t' \in \mathbb{R}^{HW \times 1}$ is a logit from the auxiliary classifier $\mathcal{H}_t$. That is, the classifier $\mathcal{H}_t$ is trained to classify all pixels in the images of $D_t$ as negatives for the next step $t+1$. Note that training images at a step $t$ might contain objects/stuff for novel classes in the future. However, we assume that noisy training signals in the step $t$ can be compensated in the future by using sufficient training images for those objects/stuff classes. Note also that the AC term is used to train the auxiliary classifier $\mathcal{H}_t$ only, and thus a feature extractor does not receive any gradients from this term.

### 3.3 Initialization

At the beginning of a step $t + 1$, we initialize our model, including a feature extractor and classifiers, using the model at the step $t$ to transfer previously learned knowledge (Fig. 1(b)). Note that classifiers for novel classes are newly added to predict logits for corresponding classes at the step $t + 1$. Note also that the number of negative samples for novel classes might not be sufficient in a dataset $D_{t+1}$. In this context, exploiting a random initialization technique for the new classifiers is not effective, degrading the discriminative ability of classifiers for the negatives. To address this, we propose to initialize new classifiers using an auxiliary classifier in a previous step, $\mathcal{H}_t$. The auxiliary classifier $\mathcal{H}_t$ contains prior knowledge of negatives for novel classes. Concretely, we initialize classifiers $\mathcal{G}_{t+1}$ in the step $t + 1$, as follows:

$$\{\mathbf{w}_{t+1}(c), \mathbf{b}_{t+1}(c)\} = \begin{cases} \{\mathbf{w}'_t, \mathbf{b}'_t\} & \text{if } c \in \mathcal{C}_{t+1} \\ \{\mathbf{w}_t(c), \mathbf{b}_t(c)\} & \text{otherwise,} \end{cases} \tag{11}$$

where we denote by $\mathbf{w}'_t$ and $\mathbf{b}'_t$ weights and a bias of the auxiliary classifier, respectively. That is, the parameters of new classifiers for novel classes $\mathcal{C}_{t+1}$ are initialized with the ones from the previous auxiliary classifier $\mathcal{H}_t$, and those for other classifiers are initialized with the counterparts from the previous step.

### 3.4 Inference

We predict pixel-level semantic labels at a step $t$, using class probabilities $\mathbf{p}_t$, as follows:

$$\hat{\mathbf{y}}(i) = \begin{cases} c_{\text{bg}} & \max_{c \in \mathcal{C}_{1:t}} \mathbf{p}_t(i, c) < \tau \\ \text{argmax}_c \, \mathbf{p}_t(i, c) & \text{otherwise,} \end{cases} \tag{12}$$

where $\tau$ is a threshold, empirically set to 0.5, and we denote by $c_{\text{bg}}$ a background class. We assign a class label to each pixel, only when one of class probabilities for the pixel is at least larger than the threshold $\tau$. Otherwise, the pixel is assigned as a background class.

## 4 Experiments

### 4.1 Implementation details

**Datasets.** We use PASCAL VOC [11] and ADE20K [30] datasets for evaluation. PASCAL VOC [11] consists of $10,582$ training and $1,449$ validation images for 20 object and background classes. ADE20K [30] provides $20,210$ and $2,000$ images for training and validation, respectively, with 150 object and stuff classes. Following the protocol in [5], we use official validation splits for evaluation. We also exclude $20\%$ of training sets, and use them to tune hyper-parameters.

**Experimental protocols.** We follow the experimental protocols in [5]. First, we evaluate our model for various incremental scenarios. Specifically, we split all object/stuff classes into base and novel ones. We train the model for base classes in an initial step, and update it sequentially for novel classes in each of the following training steps. We denote by $(N_b\text{-}N_n)$ the incremental scenario, where $N_b$ and $N_n$ are the numbers of base and novel classes, respectively. For example, given 20 object classes in PASCAL VOC [11], for an incremental scenario of (15-1), we learn 15 base classes initially, and add a single novel class sequentially, which requires 6 training steps in total. Second, we train our model under two configurations: *Disjoint* and *overlapped* settings. The disjoint setting uses a unique set of training samples for each training step. Training images in the set depict object/stuff classes belonging to one of categories to learn in a current step. The disjoint setting however excludes the images if they have any pixels regarding novel classes to be presented in the future. The overlapped setting leverages all training images that contain at least a single instance of classes to learn in a current step. Note that the overlapped setting is more realistic in practice, since the disjoint setting assumes that novel classes to learn in the future are known in advance at each training step. We perform experiments on PASCAL VOC on both disjoint and overlapped settings with incremental scenarios of (19-1), (15-5), and (15-1). For ADE20K [30], we evaluate our model under the overlapped setting only, with the scenarios of (100-50), (100-10), and (50-50), following [6, 9].

Table 1: Quantitative results on the validation split of PASCAL VOC [11] for *disjoint* and *overlapped* settings. All numbers are obtained by averaging results over five runs with standard deviations in parenthesis.

| | | 19-1 (2 steps) | | | | 15-5 (2 steps) | | | | 15-1 (6 steps) | | | |
|---|---|---|---|---|---|---|---|---|---|---|---|---|---|
| | | $mIoU_b$ | $mIoU_n$ | hIoU | $mIoU_{all}$ | $mIoU_b$ | $mIoU_n$ | hIoU | $mIoU_{all}$ | $mIoU_b$ | $mIoU_n$ | hIoU | $mIoU_{all}$ |
| Disjoint | MiB [5] | 69.60 | 25.60 | 37.43 | 67.40 | 71.80 | 43.30 | 54.02 | 64.70 | 46.20 | 12.90 | 20.17 | 37.90 |
| | SDR [22] | 69.90 | 37.30 | 48.64 | 68.40 | 73.50 | 47.30 | 57.56 | 67.20 | 59.20 | 12.90 | 21.18 | 48.10 |
| | PLOP [9] | 75.37 | 38.89 | 51.31 | 73.64 | 71.00 | 42.82 | 53.42 | 64.29 | 57.86 | 13.67 | 22.12 | 46.48 |
| | SSUL [6] | 77.38 | 22.43 | 34.78 | 74.76 | 76.44 | 45.60 | 57.12 | 69.10 | 73.97 | 32.15 | 44.82 | 64.01 |
| | RCIL [29] | - | - | - | - | 75.00 | 42.80 | 54.50 | 67.30 | 66.10 | 18.20 | 28.54 | 54.70 |
| | Ours | 77.43 | 43.56 | **55.72** | **75.81** | 77.56 | 54.13 | **63.76** | **71.98** | 76.34 | 39.36 | **51.92** | **67.54** |
| | | (±0.07) | (±2.43) | (±2.00) | (±0.15) | (±0.26) | (±0.87) | (±0.65) | (±0.36) | (±0.55) | (±2.07) | (±1.89) | (±0.82) |
| | RECALL [18] | 65.00 | 47.10 | 54.62 | 65.40 | 69.20 | 52.90 | 59.96 | 66.30 | 67.60 | 49.20 | 56.95 | 64.30 |
| | SSUL-M [6] | 77.58 | 43.89 | 56.06 | 75.98 | 76.47 | 48.55 | 59.39 | 69.83 | 76.46 | 43.37 | 55.35 | 68.58 |
| | Ours-M | 77.62 | 56.86 | **65.63** | **76.64** | 77.71 | 55.43 | **64.70** | **72.40** | 77.25 | 48.20 | **59.36** | **70.33** |
| | | (±0.12) | (±1.71) | (±1.16) | (±0.17) | (±0.21) | (±0.69) | (±0.52) | (±0.29) | (±0.20) | (±1.15) | (±0.86) | (±0.28) |
| Overlapped | MiB [5] | 70.20 | 22.10 | 33.62 | 67.80 | 75.50 | 49.40 | 59.72 | 69.00 | 35.10 | 13.50 | 19.50 | 29.70 |
| | SDR [22] | 69.10 | 32.60 | 44.30 | 67.40 | 75.40 | 52.60 | 61.97 | 69.90 | 44.70 | 21.80 | 29.31 | 39.20 |
| | PLOP [9] | 75.35 | 37.35 | 49.94 | 73.54 | 75.73 | 51.71 | 61.46 | 70.09 | 65.12 | 21.11 | 31.88 | 54.64 |
| | SSUL [6] | 77.73 | 29.68 | 42.96 | 75.44 | 77.82 | 50.10 | 60.96 | 71.22 | 77.31 | 36.59 | 49.67 | 67.61 |
| | RCIL [29] | - | - | - | - | 78.80 | 52.00 | 62.65 | 72.40 | 70.60 | 23.70 | 35.49 | 59.40 |
| | Ours | 77.76 | 41.45 | **54.03** | **76.03** | 78.83 | 58.23 | **66.98** | **73.93** | 78.09 | 42.72 | **55.21** | **69.67** |
| | | (±0.18) | (±2.91) | (±2.49) | (±0.24) | (±0.23) | (±0.45) | (±0.31) | (±0.21) | (±0.32) | (±1.58) | (±1.33) | (±0.49) |
| | RECALL [18] | 68.10 | 55.30 | 61.04 | 68.60 | 67.70 | 54.30 | 60.26 | 65.60 | 67.80 | 50.90 | 58.15 | 64.80 |
| | SSUL-M [6] | 77.83 | 49.76 | 60.71 | 76.49 | 78.40 | 55.80 | 65.20 | 73.02 | 78.36 | 49.01 | 60.30 | 71.37 |
| | Ours-M | 77.98 | 57.66 | **66.27** | **77.01** | 79.13 | 60.59 | **68.63** | **74.72** | 78.84 | 52.36 | **62.91** | **72.53** |
| | | (±0.11) | (±2.29) | (±1.51) | (±0.14) | (±0.23) | (±0.42) | (±0.25) | (±0.17) | (±0.21) | (±1.67) | (±1.22) | (±0.42) |
| | Joint | 77.57 | 77.80 | 77.68 | 77.58 | 79.48 | 71.52 | 75.29 | 77.58 | 79.48 | 71.52 | 75.29 | 77.58 |

Table 2: Quantitative results on the validation split of ADE20K [30] for an *overlapped* setting. All numbers are obtained by averaging results over five runs with standard deviations in parenthesis.

| | 100-50 (2 steps) | | | | 100-10 (6 steps) | | | | 50-50 (3 steps) | | | |
|---|---|---|---|---|---|---|---|---|---|---|---|---|
| | $mIoU_b$ | $mIoU_n$ | hIoU | $mIoU_{all}$ | $mIoU_b$ | $mIoU_n$ | hIoU | $mIoU_{all}$ | $mIoU_b$ | $mIoU_n$ | hIoU | $mIoU_{all}$ |
| MiB [5] | 40.52 | 17.17 | 24.12 | 32.79 | 38.21 | 11.12 | 17.23 | 29.24 | 45.57 | 21.01 | 28.76 | 29.31 |
| PLOP [9] | 41.87 | 14.89 | 21.97 | 32.94 | 40.48 | 13.61 | 20.37 | 31.59 | 48.83 | 20.99 | 29.36 | 30.40 |
| SSUL [6] | 41.28 | 18.02 | 25.09 | 33.58 | 40.20 | 18.75 | 25.57 | 33.10 | 48.38 | 20.15 | 28.45 | 29.56 |
| RCIL [29] | 42.30 | 18.80 | 26.03 | 34.50 | 39.30 | 17.60 | 24.31 | 32.10 | 48.30 | 25.00 | 32.95 | 32.95 |
| Ours | 42.41 | 22.89 | **29.74** | **35.95** | 41.56 | 19.51 | **26.55** | **34.26** | 48.84 | 26.28 | **34.17** | **33.90** |
| | (±0.42) | (±0.37) | (±0.40) | (±0.38) | (±0.36) | (±0.35) | (±0.32) | (±0.24) | (±0.34) | (±0.60) | (±0.52) | (±0.43) |
| SSUL-M [6] | 42.79 | 17.54 | 24.88 | 34.37 | 42.86 | 17.66 | 25.01 | 34.46 | 49.12 | 20.10 | 28.53 | 29.77 |
| Ours-M | 42.43 | 22.95 | **29.79** | **35.98** | 41.74 | 20.11 | **27.14** | **34.58** | 48.84 | 26.31 | **34.19** | **33.92** |
| | (±0.43) | (±0.36) | (±0.39) | (±0.39) | (±0.33) | (±0.27) | (±0.26) | (±0.25) | (±0.28) | (±0.59) | (±0.51) | (±0.41) |
| Joint | 43.16 | 30.03 | 35.42 | 38.81 | 43.16 | 30.03 | 35.42 | 38.81 | 49.35 | 33.44 | 39.86 | 38.81 |

**Training.** We use the DeepLabV3 [7] architecture using ResNet-101 [13] pretrained on ImageNet [26] as a backbone network. Following [5, 6, 9, 22], we adopt different training strategies for each dataset. For PASCAL VOC [11], we train our model with 60 epochs for both initial and incremental steps, with a batch size of 32. We adopt a polynomial learning rate scheduler, where learning rates are set to 0.001 and 0.0001 for initial and incremental steps, respectively. We empirically set $\gamma$ to 2 during an initial step and 1 for others. For ADE20K [30], we train our model for 100 epochs with a batch size of 24. Following [6], we adopt the poly learning rate scheduler with a linear warm-up [12], where learning rates are set to 0.0025 for an initial step and 0.00025 for incremental ones, respectively. We set $\gamma$ to 35 for all training steps. For both datasets, we adopt the SGD optimizer with momentum of 0.9, and set $\alpha$ and $\beta$ to 5. We implement our model using PyTorch [24] and train it with four NVIDIA RTX A5000 GPUs.

**Evaluation metrics.** Following [5, 6, 9, 22], we report $mIoU_b$, $mIoU_n$, and $mIoU_{all}$ scores, that is, intersection-over-union (IoU) scores averaged over base, novel and all classes, respectively. Simply averaging the IoU score over all classes (*i.e.*, $mIoU_{all}$) is not appropriate to evaluate the performance of CISS models, especially for the case that the number of novel classes is relatively small compared to that of base ones. Accordingly, we also report a harmonic mean (hIoU) of $mIoU_b$ and $mIoU_n$ scores, which is less susceptible to the imbalance between base and novel classes.

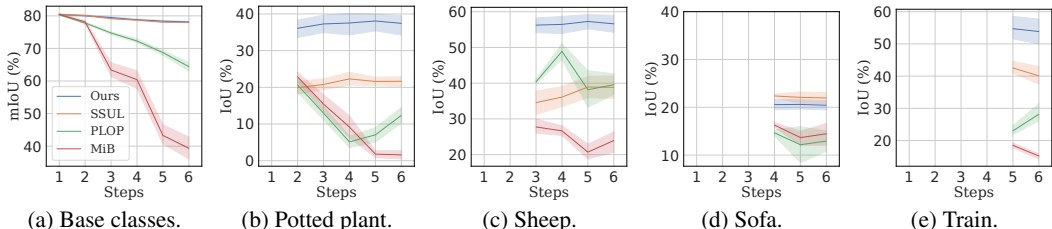

| (a) Base classes. | (b) Potted plant. | (c) Sheep. | (d) Sofa. | (e) Train. |

Figure 3: mIoU comparisons of state-of-the-art CISS methods [5, 6, 9] over training steps. We train CISS models for 15 base classes initially, and add a single novel class for each incremental step (*i.e.*, 15-1 setting with 6 steps). We show variations of mIoU scores for base classes (a) and for individual novel classes separately (b-e).

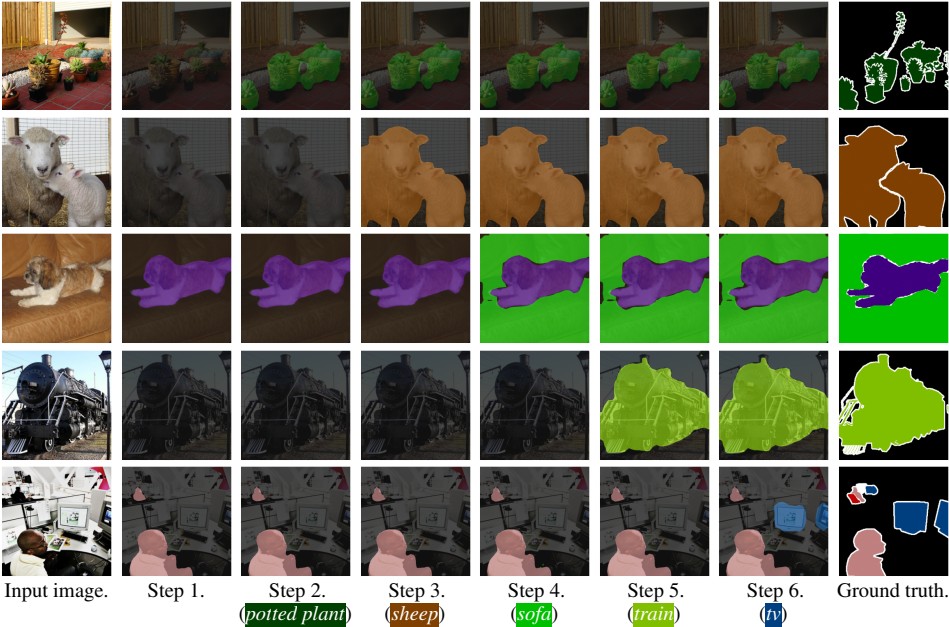

| Input image. | Step 1. | Step 2. (*potted plant*) | Step 3. (*sheep*) | Step 4. (*sofa*) | Step 5. (*train*) | Step 6. (*tv*) | Ground truth. |

Figure 4: Qualitative results for the 15-1 *overlapped* setting on PASCAL VOC [11]. *chair*, *dog*, and *person* belong to base classes.

## 4.2 Results

We show in Tables 1 and 2 quantitative comparisons between ours and state-of-the-art CISS methods [5, 6, 9, 22] on PASCAL VOC [11] and ADE20K [30], respectively. To better demonstrate the effectiveness of our approach, we also report results for joint training that serve as upper bounds. For fair comparison with SSUL-M [6] exploiting an external memory, we also report the results (Ours-M) obtained using previous training samples, following the official implementation provided by the authors.

We can observe from Tables 1 and 2 three things: (1) CISS methods using a BCE term [6] to learn novel classes, including ours, perform better than other approaches [5, 9, 22] employing a softmax CE term, demonstrating the drawback of the softmax function for CISS. (2) Among competitive methods without using an external memory, our method achieves a new state of the art for all cases in terms of mIoU$_{all}$ and hIoU scores. This suggests that ours preserves knowledge learned from base classes, while learning novel ones effectively, compared to other methods, achieving a better compromise between rigidity and plasticity for CISS. We can also see that our method outperforms others in terms of a hIoU score for all cases by a significant margin. (3) The external memory provides complementary information, and this brings additional performance gains. Our method (Ours-M) clearly outperforms SSUL-M in terms of mIoU$_{all}$ and hIoU scores. Note that SSUL and SSUL-M also exploit an off-the-shelf saliency detector [15], pretrained on additional training samples [17], which requires more computational power and memory for training.

Table 3: Quantitative comparisons for variants of our method under the *overlapped* setting on PASCAL VOC [11]. All numbers are obtained by averaging results over five runs with standard deviations.

| Baseline ($\mathcal{L}_{mbce} + \mathcal{L}_{kd}$) | Initialization | | $\mathcal{L}_{dkd}$ | | 15-1 (6 steps) | | | |
|:---:|:---:|:---:|:---:|:---:|:---:|:---:|:---:|:---:|
| | Random | Ours | $\mathcal{L}_{dkd}^+$ | $\mathcal{L}_{dkd}^-$ | mIoU$_b$ | mIoU$_n$ | hIoU | mIoU$_{all}$ |
| ✓ | ✓ | | | | 76.04±0.82 | 35.16±1.53 | 48.07±1.48 | 66.30±0.83 |
| ✓ | ✓ | | ✓ | ✓ | 77.97±0.32 | 36.53±1.18 | 49.74±1.10 | 68.10±0.42 |
| ✓ | | ✓ | | | 74.43±1.15 | 39.41±1.51 | 51.53±1.53 | 66.09±1.19 |
| ✓ | | ✓ | ✓ | | 77.94±0.35 | 42.47±1.54 | 54.80±1.31 | 69.45±0.51 |
| ✓ | | ✓ | | ✓ | 75.92±1.00 | 40.27±1.46 | 52.62±1.42 | 67.43±1.04 |
| ✓ | | ✓ | ✓ | ✓ | 78.09±0.32 | 42.72±1.58 | **55.21**±1.33 | **69.67**±0.49 |

(a) $\|\mathbf{z}_t^+ - \mathbf{z}_{t-1}^+\|$.    (b) $\|\mathbf{z}_t^- - \mathbf{z}_{t-1}^-\|$.    (c) $\|\mathbf{z}_t - \mathbf{z}_{t-1}\|$.

Figure 5: Quantitative comparisons of our model trained with and without the DKD term. We plot values of $\|\mathbf{z}_t^+ - \mathbf{z}_{t-1}^+\|$, $\|\mathbf{z}_t^- - \mathbf{z}_{t-1}^-\|$, and $\|\mathbf{z}_t - \mathbf{z}_{t-1}\|$ over iterations. We obtain the results for the single incremental step under a 19-1 scenario on PASCAL VOC [11].

We also compare in Fig. 3 our method with the state of the art, including MiB [5], PLOP [9] and SSUL [6], over training steps in terms of mIoU. We can see that our method avoids catastrophic forgetting effectively, maintaining mIoU scores for both base and novel classes over a number of steps. In contrast, other methods often fail to preserve the mIoU scores of old classes in later steps. Except for the novel class at the fourth incremental step in Fig. 3(d), where SSUL [6] is slightly better than ours, our approach outperforms the state of the art for all training steps by a significant margin.

We show in Fig. 4 qualitative results on the PASCAL VOC [11]. We can see that our method successfully learns novel classes incrementally without forgetting previously learned classes. More qualitative results can be found in the supplement.

## 4.3 Discussion

We show in Table 3 an ablation analysis of our approach. The baseline model in the first row uses mBCE and KD terms only. Note that the baseline already performs comparable to state-of-the-art methods [5, 6, 9, 22], confirming once more the significance of employing the BCE loss for CISS. The last row shows that exploiting the DKD loss along with our initialization technique achieves the best performance for all metrics. We further provide a detailed analysis for the two components in the following.

**DKD.** From the first and the second rows in Table 3, we can see that the DKD term boosts the performance for both base and novel classes, as it helps to retain knowledge even after incremental steps. Moreover, the second row shows that adopting the DKD term performs significantly better, compared to freezing a feature extractor as in SSUL [6], in terms of mIoU$_b$ score (See the result in Table 1). This suggests that our DKD technique is more effective for maintaining the rigidity of a CISS model. We plot in Fig. 5 numerical values of $\|\mathbf{z}_t^+ - \mathbf{z}_{t-1}^+\|$, $\|\mathbf{z}_t^- - \mathbf{z}_{t-1}^-\|$, and $\|\mathbf{z}_t - \mathbf{z}_{t-1}\|$ over iterations, where $\|\cdot\|$ is the L2 norm. As positive and negative reasoning scores, $\mathbf{z}_t^+$ and $\mathbf{z}_t^-$, are similar to previous ones, $\mathbf{z}_{t-1}^+$ and $\mathbf{z}_{t-1}^-$, respectively, $\|\mathbf{z}_t^+ - \mathbf{z}_{t-1}^+\|$ and $\|\mathbf{z}_t^- - \mathbf{z}_{t-1}^-\|$ becomes smaller. We can see from Figs. 5(a) and (b) that the model trained without the DKD term does not preserve the reasoning scores effectively. On the contrary, the DKD term prevents abrupt changes of the scores. Note that $\|\mathbf{z}_t - \mathbf{z}_{t-1}\|$ quantifies the consistency of predictions before and after an incremental step, suggesting that minimizing $\|\mathbf{z}_t - \mathbf{z}_{t-1}\|$ is also crucial to alleviate catastrophic forgetting. We can observe in Fig. 5(c) that the DKD term helps to further minimize $\|\mathbf{z}_t - \mathbf{z}_{t-1}\|$. These results verify that the DKD term improves the rigidity of a CISS model together with KD.

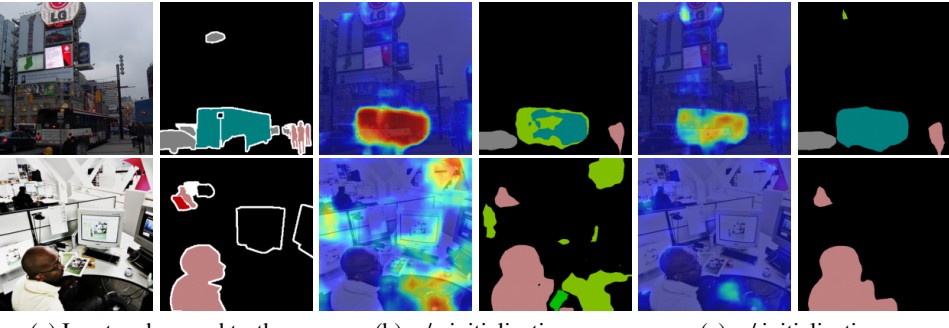

| (a) Input and ground truth. | (b) w/o initialization. | (c) w/ initialization. |

Figure 6: Visual comparisons of activation maps and segmentation results for the 15-1 *overlapped* setting of PASCAL VOC [11]. We show activation maps for *train* and predictions using our model with and without the initialization technique. Our model learns a *train* class in the 4-th incremental step, after learning *potted plant*, *sheep*, and *sofa* incrementally. *car*, *bus*, *person*, *chair* belong to base classes.

**Initialization.**    The first and third rows in Table 3 demonstrate that our initialization technique itself provides a considerable performance gain for novel classes in terms of the $mIoU_n$ score, verifying that properly initializing classifier weights for novel classes is crucial for CISS. The initialization technique alleviates the problem, caused by a lack of negatives at each incremental step, and provides strong prior knowledge to learn novel classes. Note that our model in the third row already gives competitive results compared to SSUL [6] in terms of the $mIoU_n$ score, even without exploiting an off-the-shelf saliency detector [15]. We provide in Fig. 6 visual comparisons of activation maps for a novel class (*i.e.*, *train*), and segmentation labels for all target classes (*i.e.*, 15 base classes and the incremental ones of *potted plant*, *sheep*, *sofa*, *train*). We obtain the results using classifiers trained with and without our initialization technique. We can see that the classifiers without our initialization often activates incorrectly on background regions (Fig. 6(b) bottom) or previous classes (*i.e.*, *bus* in Fig. 6(b) top). This distracts classifiers for previous classes, resulting in incorrect semantic labels in the regions. On the contrary, the classifiers initialized with our technique successfully suppresses false class probabilities for those regions, providing better segmentation results.

Note that we train an auxiliary classifier in advance of learning novel classes for incremental steps. This suggests that the initialization technique can be adopted only when segmentation models are trained from scratch, and thus pre-trained off-the-shelf segmentation models could not be exploited directly. Other techniques including mBCE and DKD in Sec. 3.2 can be applied to the off-the-shelf models, which brings significant performance gains, compared to current CISS methods [5, 9, 22, 29] (See Tables 1 and 3).

Note also that the auxiliary classifier makes segmentation models aware of the fact that novel classes will be added in the future. This increment-aware continual learning achieves a new state of the art on standard CISS benchmarks with negligible computational overhead for training, and it will provide a novel paradigm for CIL.

## 5    Conclusion

We have presented a novel framework that shows a good trade-off between rigidity and plasticity for class-incremental semantic segmentation (CISS). In particular, we have introduced a new learning paradigm, decompose to distill knowledge, to improve the rigidity, and have proposed a novel initialization technique to learn novel classes better. Finally, we have shown that our framework achieves a new state of the art on standard CISS benchmarks.

## Acknowledgments and Disclosure of Funding

This work was partly supported by Institute of Information & communications Technology Planning & Evaluation (IITP) grant funded by the Korea government (MSIT) (No.RS-2022-00143524, Development of Fundamental Technology and Integrated Solution for Next-Generation Automatic Artificial Intelligence System) and the KIST Institutional Program (Project No.2E31051-21-203).

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
