# Decomposed Knowledge Distillation for Class-Incremental Semantic Segmentation Supplement

Donghyeon Baek[1]        Youngmin Oh[1]        Sanghoon Lee[1]
Junghyup Lee[1]        Bumsub Ham[1,2*]

[1]Yonsei University        [2]Korea Institute of Science and Technology (KIST)

https://cvlab.yonsei.ac.kr/projects/DKD/

## S1  More results

### S1.1  More quantitative results

Table S1: Quantitative results on the validation split of PASCAL VOC [4] for an *overlapped* setting. All numbers are obtained by averaging results over five runs with standard deviations in parenthesis. †: results are taken from [2].

|  | 10-1 (11 steps) | | | | 5-3 (6 steps) | | | |
|---|---|---|---|---|---|---|---|---|
|  | $mIoU_b$ | $mIoU_n$ | hIoU | $mIoU_{all}$ | $mIoU_b$ | $mIoU_n$ | hIoU | $mIoU_{all}$ |
| MiB† [1] | 12.25 | 13.09 | 12.66 | 12.65 | 57.10 | 42.56 | 48.77 | 46.71 |
| PLOP† [3] | 44.03 | 15.51 | 22.94 | 30.45 | 17.48 | 19.16 | 18.28 | 18.68 |
| SSUL [2] | 71.31 | 45.98 | 55.91 | 59.25 | 72.44 | 50.67 | 59.63 | 56.89 |
| RCIL [6] | 55.40 | 15.10 | 23.73 | 34.30 | - | - | - | - |
| Ours | 73.10 | 46.51 | **56.84** | **60.44** | 69.57 | 53.52 | **60.49** | **58.10** |
|  | ($\pm$0.56) | ($\pm$1.27) | ($\pm$0.97) | ($\pm$0.69) | ($\pm$0.36) | ($\pm$0.91) | ($\pm$0.57) | ($\pm$0.64) |
| RECALL [5] | 65.00 | 53.70 | 58.81 | 60.70 | - | - | - | - |
| SSUL-M [2] | 74.02 | 53.23 | 61.93 | 64.12 | 71.27 | 53.21 | 60.93 | 58.37 |
| Ours-M | 74.04 | 56.67 | **64.19** | **65.77** | 69.77 | 60.21 | **64.63** | **62.94** |
|  | ($\pm$0.40) | ($\pm$1.72) | ($\pm$1.15) | ($\pm$0.88) | ($\pm$0.46) | ($\pm$0.76) | ($\pm$0.34) | ($\pm$0.46) |
| Joint | 78.51 | 76.56 | 77.52 | 77.58 | 77.40 | 77.65 | 77.53 | 77.58 |

We provide in Table S1 an additional quantitative comparison between ours and state-of-the-art CISS methods [1, 2, 3] on PASCAL VOC [4]. The models are trained under an *overlapped* setting with incremental scenarios of (10-1) and (5-3). We can see that our method achieves a new state of the art in terms of both $mIoU_{all}$ and hIoU scores, confirming once more the effectiveness of our framework.

### S1.2  Per-class results

Table S2: Per-class IoU scores on the validation split of PASCAL VOC [4] for an *overlapped* setting.

|  |  | bg. | aero | bike | bird | boat | bott | bus | car | cat | chair | cow | table | dog | horse | mbik | persn | plnt | shee | sofa | train | tv | $mIoU_{all}$ |
|---|---|---|---|---|---|---|---|---|---|---|---|---|---|---|---|---|---|---|---|---|---|---|---|
| 19-1 | SSUL [2] | 92.2 | **88.9** | **40.5** | **89.6** | **70.9** | 79.6 | **95.1** | 88.6 | **93.7** | 37.3 | 84.8 | 59.8 | 89.5 | **85.9** | **87.6** | 84.6 | 61.3 | 82.3 | **54.5** | **88.1** | 29.7 | 75.4 |
| 19-1 | Ours | **92.4** | 87.2 | 40.2 | 87.8 | 67.6 | **81.6** | 93.2 | **90.8** | 92.7 | **37.5** | **86.2** | **63.2** | **89.9** | 85.4 | 85.2 | **86.4** | **66.5** | **83.1** | 49.6 | 86.9 | **45.0** | **76.1** |
| 15-5 | SSUL [2] | **91.5** | **89.8** | 40.0 | 88.6 | **70.3** | **81.2** | 89.7 | 88.0 | 92.8 | **36.8** | 75.7 | 56.7 | **90.0** | 83.6 | **85.2** | 85.4 | 36.0 | 57.7 | 32.1 | 70.1 | 54.6 | 71.2 |
| 15-5 | Ours | 91.1 | 88.4 | **40.7** | **89.6** | 68.6 | 81.1 | **92.1** | **88.6** | **93.2** | 36.7 | **83.2** | **62.4** | 89.2 | **84.5** | 84.8 | **86.2** | **44.2** | **69.5** | **33.0** | **80.9** | **65.1** | **74.0** |
| 15-1 | SSUL [2] | **89.6** | 89.0 | **41.0** | 88.4 | 69.5 | 80.8 | 85.6 | 88.5 | 92.6 | 35.5 | 77.0 | 56.7 | **90.1** | 83.5 | 84.3 | 85.1 | 33.0 | 49.6 | **26.6** | 43.4 | 30.4 | 67.6 |
| 15-1 | Ours | 87.4 | **89.3** | 40.7 | **88.8** | **70.1** | 79.9 | **90.6** | **89.0** | **93.1** | **37.1** | **80.1** | **62.7** | 89.3 | **84.8** | **84.4** | **86.2** | **38.0** | **59.4** | 20.9 | **62.0** | **46.5** | **70.5** |

We show in Table S2 per-class IoU scores on PASCAL VOC [4] for an *overlapped* setting. We can clearly see that our method outperforms SSUL [2], especially for *train* and *sheep* classes in the (15-5)

---

*Corresponding author

36th Conference on Neural Information Processing Systems (NeurIPS 2022).

and (15-1) scenarios. They belong to novel classes in those scenarios, and have similar appearance or context with one of the base classes. For example, a *train* class has similar appearance with a *bus* class, and a *sheep* class has similar context with a *cow* class. This implies that SSUL, which freezes a feature extractor, struggles with differentiating visually or contextually similar classes, while our model learns discriminative features for novel classes, enabling distinguishing the similar classes.

## S1.3   Additional analysis for initialization technique

Table S3: Quantitative comparison for variants of our method under the overlapped setting on PASCAL VOC [4]. All numbers are obtained by averaging results over five runs with standard deviations.

| Baseline $(\mathcal{L}_{mbce} + \mathcal{L}_{kd})$ | Initialization | | $\mathcal{L}_{dkd}$ | 19-1 (2 steps) | | | |
|---|---|---|---|---|---|---|---|
| | Random | Ours | | $mIoU_b$ | $mIoU_n$ | hIoU | $mIoU_{all}$ |
| ✓ | ✓ | | ✓ | 77.89±0.10 | 32.44±5.42 | 45.59±5.58 | 75.73±0.28 |
| ✓ | | ✓ | ✓ | 77.76±0.18 | 41.45±2.91 | **54.03**±2.49 | **76.03**±0.24 |

We show in Table S3 additional experimental results for our method with and without the initialization technique under the (19-1) overlapped setting on PASCAL VOC [11]. All numbers are obtained by averaging results over five runs with standard deviations. From Table R1, we can see that applying the initialization technique gives a large $mIoU_n$ gain of 9.01%, confirming once more the effectiveness of the initialization technique.

Table S4: Quantitative comparison for variants of our method under the overlapped setting on PASCAL VOC [4]. All numbers are obtained by averaging results over five runs with standard deviations.

| Baseline $(\mathcal{L}_{mbce} + \mathcal{L}_{kd})$ | Initialization | | $\mathcal{L}_{dkd}$ | 19-1 (2 steps) | |
|---|---|---|---|---|---|
| | Random | Ours | | Recall | Precision |
| ✓ | ✓ | | ✓ | 78.88±0.93 | 35.60±6.41 |
| ✓ | | ✓ | ✓ | 78.40±1.06 | **46.80**±3.43 |

To further analyze the effectiveness of the initialization technique, we present in Table S4 recall and precision scores for a tv class, which belongs to a novel class in the (19-1) overlapped setting on PASCAL VOC. All numbers are also obtained by averaging results over five runs with standard deviations. Recall and precision are measured by $\frac{N_{TP}}{N_{TP}+N_{FN}}$ and $\frac{N_{TP}}{N_{TP}+N_{FP}}$, respectively, where $N_{TP}$, $N_{FP}$, and $N_{FN}$ are the number of true positives, false positives, and false negatives, respectively. We can see that our methods with and without the initialization technique show similar recall scores, suggesting that both classify tv objects well as a tv class, even without the initialization technique.

On the other hand, our method with the initialization technique outperforms its counterpart in terms of the precision score. This indicates that the initialization technique reduces the number of false positives and explains the reason why the initialization technique boosts the $mIoU_n$ scores in Table S3. These results demonstrate once more the effectiveness of our initialization technique that allows a novel classifier to learn from abundant negative samples in previous learning steps, improving the discriminative ability of the novel classifier.

## S1.4   Hyperparameter analysis

Table S5: Quantitative comparison for variants of the value of $\alpha$ under the overlapped setting on PASCAL VOC [4]. All numbers are obtained by averaging results over five runs with standard deviations.

| $\alpha$ | 19-1 (2 steps) | | | |
|---|---|---|---|---|
| | $mIoU_b$ | $mIoU_n$ | hIoU | $mIoU_{all}$ |
| 0 | 21.31±3.43 | 2.27±1.88 | 4.11±3.07 | 16.78±2.86 |
| 1 | 60.13±4.75 | 28.28±6.20 | 38.47±6.68 | 52.55±4.96 |
| 5 | 74.43±1.15 | 39.41±1.51 | **51.53**±1.53 | **66.09**±1.19 |
| 10 | 66.64±8.10 | 36.36±2.33 | 46.97±3.85 | 59.40±6.66 |

We fix $\beta$ to 0, and vary the value of $\alpha$ within {0,1,5,10}. We show in Table S5 results for different values of $\alpha$ under the (15-1) overlapped setting on PASCAL VOC [4]. All numbers are obtained by averaging results over five runs with standard deviations. From the first row in Table R3, we can see that training a CISS model without using both the KD and DKD terms in Eq. (1) causes catastrophic forgetting, drastically deteriorating the performance. We can see that employing KD is particularly effective in alleviating the forgetting problem. We have empirically set $\alpha$ to 5 for all experiments.

Table S6: Quantitative comparison for variants of the value of $\beta$ under the overlapped setting on PASCAL VOC [4]. All numbers are obtained by averaging results over five runs with standard deviations.

| $\beta$ | 19-1 (2 steps) | | | |
| --- | --- | --- | --- | --- |
| | $mIoU_b$ | $mIoU_n$ | hIoU | $mIoU_{all}$ |
| 0 | 74.43±1.15 | 39.41±1.51 | 51.53±1.53 | 66.09±1.19 |
| 1 | 77.57±0.57 | 42.14±1.39 | 54.62±1.20 | 69.14±0.59 |
| 5 | 78.09±0.32 | 42.72±1.58 | **55.23**±1.33 | **69.67**±0.49 |
| 10 | 76.93±0.85 | 41.30±2.32 | 53.75±2.08 | 68.45±1.04 |

We also vary the value of $\beta$ between {0,1,5,10} while $\alpha$ is set to 5. We present in Table S6 results for different values of $\beta$ under the overlapped setting with the scenarios of (15-1) on PASCAL VOC. All numbers are obtained by averaging results over five runs with standard deviations. We can see that setting $\beta$ to be a positive value always provides better results. This validates the effectiveness of the DKD term. We can also see from Table S6 that our method is robust to various choices of $\beta$.

## S1.5 More Qualitative results

We present in Fig. S1 qualitative results on ADE20K [7]. We can also see that our method is able to learn novel classes incrementally without forgetting previously learned classes.

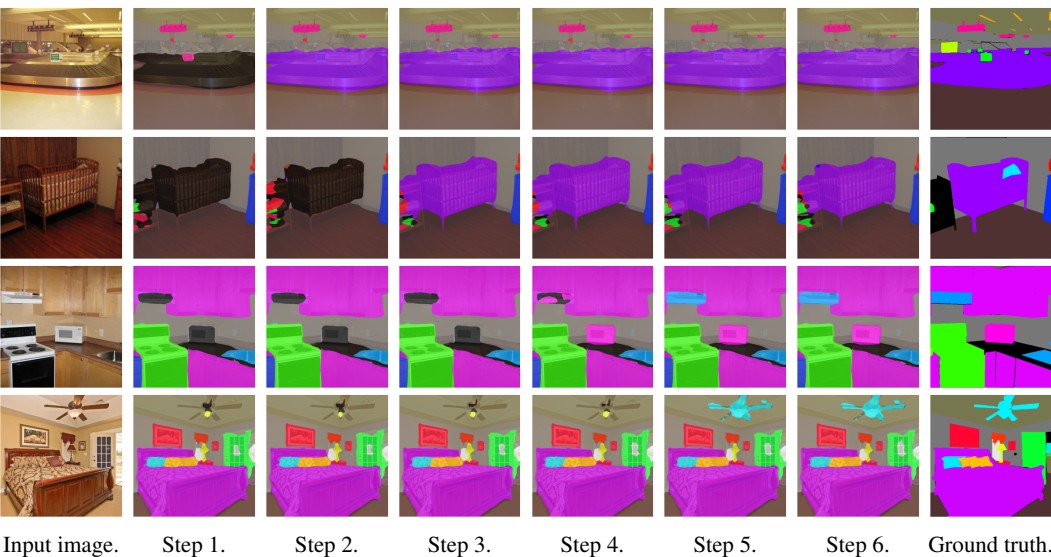

| Input image. | Step 1. | Step 2. | Step 3. | Step 4. | Step 5. | Step 6. | Ground truth. |

Figure S1: Qualitative results for the 100-10 *overlapped* setting on ADE20K [7]. *conveyer belt*, *cradle*, *microwave*, *hood*, and *fan* belong to novel classes.