# OpenReview forum: "Decomposed Knowledge Distillation for Class-Incremental Semantic Segmentation"
_NeurIPS.cc/2022/Conference — NeurIPS 2022 Accept_

### Official Review · Reviewer_X4eN · 2022-07-04

**Rating:** 6
**Confidence:** 5
**Soundness:** 2 fair
**Presentation:** 4 excellent
**Contribution:** 3 good

**Summary:**

The paper proposes a technique for class-incremental learning in semantic segmentation. First, a knowledge distillation loss is proposed, decomposing positive and negative class logits into separate loss functions. Second, an initialization strategy for the classifiers of novel classes is proposed, which is relies on an auxiliary classifier. Evaluation on several standard benchmarks for incremental semantic segmentation shows the methods effectiveness.

**Questions:**

1. Could you state from where you adopted the knowledge distillation loss in Eq. 5? Also from SSUL [6]?

2. Do the 5 different runs use different random seeds?


**Limitations:**

Limitations have been discussed to some degree in the supplementary. I think there are, however, two limitations which have not been clearly stated. The method requires modifications to the semantic segmentation model, which makes the method not applicable to off-the-shelf models (cf. page 5, ll. 181-185). Also, the assumption that new classes rarely appear in in old data (page 5, ll. 186-188) is true for Pascal VOC, but not for more complex scenes such as Cityscapes, where such classes would often just not be labeled. This might make this method rather applicable to simple scenes. I would be happy to hear the authors’ take on these thoughts.

**Strengths And Weaknesses:**

Strengths:

1. The introduction and related work provide a clear motivation for the proposed method and overall discuss the contributions well w.r.t. to related works.

2. Clear and extensive mathematical description of the method, which seems reproducible, although code is not provided in the submission.

3. State-of-the-art results on the standard incremental semantic segmentation benchmarks.

4. Nicely prepared figures and tables, clarity of the overall manuscript.

Weaknesses:

1. Lack of hyperparameter analysis: The methodology on how hyperparameters are determined is well-described, but I did not find an analysis in this work, on how changes in the method’s hyperparameters affect the performance.

2. Applicability: The method requires a modification to the pre-trained model before the incremental steps. Therefore, the method is not applicable to off-the-shelf semantic segmentation models and from an applicability perspective rather comparable to replay-based approaches. I still think, the memory-footprint of the method is much smaller than other replay-based methods (e.g., SSUL), but I think the limitation should be clearly stated as it is a bit different from the standard incremental setting for semantic segmentation.

3. Root cause of improved performance: It is a bit unclear how much of the improvement towards SOTA performance is due to better chosen hyperparameters (such as, e.g., larger batch size on better hardware) or how much is due to the contributions. This could be easily fixed by adding a self-simulated version of MiB [5] or SSUL-M [6] in the author’s simulation setting to the comparison in Tables 1 and 2, showing how the self-simulated version of these methods compare to the original numbers provided in the papers.

Overall, I really like the approach taken in this work, the paper is very well written and although some weaknesses definitely exist, the overall value of the contributions has been shown. I would welcome a clear answer to the weaknesses in the rebuttal as some of them might be easy to address.

Minor comments and suggestions:

4. I think the related work Maracani et al. “RECALL: Replay-based Continual Learning in Semantic Segmentation,” ICCV 2021, should be discussed and added to the experimental comparison. The work is similar to the strongly related work SSUL [6] and therefore appears to be relevant also for this work.

5. The conclusion mainly repeats the abstract. It would be beneficial to put a bit more focus on future works, larger impact of this method on the field of semantic segmentation, or limitations of the method.

---

> ### Author Response · Authors · 2022-08-02
> **Response to Reviewer X4eN [3/3]**
>
> **5. Conclusion**
>
> We appreciate your thoughtful recommendations. We will add a discussion about future work and limitations including the additional ones discussed in the review.
> ***
> **6. The knowledge distillation (KD) loss**
>
> We have adopted KD loss from iCaRL [24], which also exploits the multiple binary cross-entropy loss in class-incremental image classification.
> ***
> **7. About random seeds**
>
> We have used different random seeds for the different runs.
> ***
> **8. Applicability of proposed initialization technique**
>
> We agree that our explanation (L186-188) might not hold for complex scenes. We will tone down the claim. However, it is not a critical problem for common class-incremental scenarios, where sufficient training images at an incremental step $t$ contain objects or stuff for novel classes $\mathcal{C}\_{t}$. This facilitates compensating for noisy training signals in previous learning steps. We will clarify this issue.
> ***
> References
> - [A] RECALL: Replay-based continual learning in semantic segmentation, In ICCV, 2021.
> - [B] Large scale GAN training for high fidelity natural image synthesis. In ICLR, 2019.
> - [C] Representation Compensation Networks for Continual Semantic Segmentation, In CVPR, 2022.

---

> ### Author Response · Authors · 2022-08-02
> **Response to Reviewer X4eN [2/3]**
>
> **3. Results of the self-simulated version of other works**
>
> We provide in Table R7 results for the self-simulated counterparts [5, 6, 9] under the (15-1) overlapped setting on PASCAL VOC [11]. All numbers are obtained by averaging results over five runs with standard deviations. We have followed the official implementations provided by the authors. Specifically, we train MiB [5] and PLOP [9] with a batch size of 24, and SSUL [6] with a batch size of 32. Note that we train our method with a batch size of 32. We can see that the numbers in Tables 1 and R7 are comparable, confirming that our self-simulated counterparts have been well implemented. While the works of [5, 9] and ours employ a single convolutional layer as a classifier, SSUL [6] employs a convolutional block that consists of Conv(3x3)-BN-ReLU-Conv(1x1) layers. Therefore, we also report the results for the SSUL variants that use the same classifier as other methods and its variant exploiting an external memory, denoted by $\text{SSUL}^{\dagger}$ and $\text{SSUL-M}^{\dagger}$, respectively, for fair comparison. We can clearly see that ours achieves a new state of the art in terms of $\text{mIoU}\_{\text{all}}$ and $\text{hIoU}$ scores among competitive methods without using external memory. We can also see that ours-M clearly outperforms SSUL-M in terms of $\text{mIoU}\_{\text{all}}$ and $\text{hIoU}$ scores. We will also report results for the self-simulated MiB and PLOP trained with a batch size of 32.
>
> \
> Table R7: Quantitative results on PASCAL VOC [11] for (15-1) overlapped settings.
> | | **$\text{mIoU}_{\text{b}}$**  | $\text{mIoU}_{\text{n}}$ | $\text{hIoU}$ |  $\text{mIoU}_{\text{all}}$ |
> |-|:-:|:-:|:-:|:-:|
> | $\text{MiB}$  | 39.33 $\pm$ 4.19 | 14.72 $\pm$ 0.67 | 21.39 $\pm$ 1.14 | 33.47 $\pm$ 3.27 |
> | $\text{PLOP}$ | 61.53 $\pm$ 5.42 | 17.45 $\pm$ 3.44 | 26.89 $\pm$ 4.74 | 51.48 $\pm$ 4.88 |
> | $\text{SSUL}$ | 77.78 $\pm$ 0.19 | 39.18 $\pm$ 1.12 | 52.10 $\pm$ 0.99 | 68.59 $\pm$ 0.29 |
> | $\text{SSUL}^{\dagger}$ | 78.00 $\pm$ 0.07 | 29.52 $\pm$ 2.44 | 42.78 $\pm$ 2.58 | 66.46 $\pm$ 0.58 |
> | $\text{Ours}$ | 78.09 $\pm$ 0.32 | 42.72 $\pm$ 1.58 | **55.21 $\pm$ 1.33** | **69.67 $\pm$ 0.49** |
> ||
> | $\text{SSUL-M}^{\dagger}$ | 79.15 $\pm$ 0.17 | 42.73 $\pm$ 2.52 | 55.46 $\pm$ 2.16 | 70.48 $\pm$ 0.57 |
> | $\text{Ours-M}$ | 78.84 $\pm$ 0.21 | 52.32 $\pm$ 1.65 | **62.89 $\pm$ 1.20** | **72.52 $\pm$ 0.42** |
> ***
> **4. Comparison with RECALL**
>
> We thank you for suggesting an additional reference. We show in Table R8 comparisons between ours and RECALL [A] on PASCAL VOC [11]. Note that RECALL does not report results on ADE20K [28]. Note also that RECALL exploits additional web-crawled images or pre-trained GAN [B], suggesting that we compare RECALL with a variant of our method exploiting external memory for fair comparison.  We can see that ours with an external memory outperforms RECALL for all class-incremental semantic segmentation benchmarks, even with less memory footprint required for training. We will add the results of RECALL in Table 2.
>
> \
> Table R8: Quantitative results on PASCAL VOC [11] for overlapped settings.
> | 19-1 (2 steps) | $\text{mIoU}_{\text{b}}$  | $\text{mIoU}_{\text{n}}$ | $\text{hIoU}$ |  $\text{mIoU}_{\text{all}}$ |
> |-|:-:|:-:|:-:|:-:|
> | $\text{RECALL (GAN)}$| 67.90 | 53.50 | 59.85 | 68.40 |
> | $\text{RECALL (Web)}$| 68.10 | 55.30 | 61.04 | 68.60 |
> | $\text{Ours}$ | 77.76 $\pm$ 0.18 | 41.45 $\pm$ 2.91 | 54.03 $\pm$ 2.49 | 76.03 $\pm$ 0.24 |
> | $\text{Ours-M}$ | 77.98 $\pm$ 0.11 | 57.66 $\pm$ 2.29 | **66.27 $\pm$ 1.51** | **77.01 $\pm$ 0.14** |
>
> | 15-5 (2 steps) | $\text{mIoU}_{\text{b}}$  | $\text{mIoU}_{\text{n}}$ | $\text{hIoU}$ |  $\text{mIoU}_{\text{all}}$ |
> |-|:-:|:-:|:-:|:-:|
> | $\text{RECALL (GAN)}$| 66.60 | 50.90 | 57.70 | 64.00 |
> | $\text{RECALL (Web)}$| 67.70 | 54.30 | 60.26 | 65.60 |
> | $\text{Ours}$ | 78.83 $\pm$ 0.23 | 58.23 $\pm$ 0.45 | 66.98 $\pm$ 0.31 | 73.93 $\pm$ 0.21 |
> | $\text{Ours-M}$ | 79.13 $\pm$ 0.23 | 60.59 $\pm$ 0.42 | **68.63 $\pm$ 0.25** | **74.72 $\pm$ 0.17** |
>
> | 15-1 (6 steps) | $\text{mIoU}_{\text{b}}$  | $\text{mIoU}_{\text{n}}$ | $\text{hIoU}$ |  $\text{mIoU}_{\text{all}}$ |
> |-|:-:|:-:|:-:|:-:|
> | $\text{RECALL (GAN)}$| 65.70 | 47.80 | 55.34 | 62.70 |
> | $\text{RECALL (Web)}$| 67.80 | 50.90 | 58.15 | 64.80 |
> | $\text{Ours}$ | 78.09 $\pm$ 0.32 | 42.72 $\pm$ 1.58 | 55.21 $\pm$ 1.33 | 69.67 $\pm$ 0.49 |
> | $\text{Ours-M}$ | 78.84 $\pm$ 0.21 | 52.32 $\pm$ 1.65 | **62.89 $\pm$ 1.20** | **72.52 $\pm$ 0.42** |

---

> > ### Comment · Reviewer_X4eN · 2022-08-08
> > **Reply to Author Rebuttal**
> >
> > Thank you very much for your detailed reply. Just to confirm (maybe you somehow mentioned it): Did you run the baselines in your own codebase, taking just the baseline losses (e.g., for MiB that would be very straightforward) and are the results you show in Table R7 obtained from these experiments?

---

> > > ### Author Response · Authors · 2022-08-09
> > > **Response to Reviewer X4eN**
> > >
> > > Thank you for your reply. We have obtained the results in Table R7 using the official code provided by the authors of MiB [5], PLOP [9], and SSUL [6]. We further integrate MiB into our own codebase to compare the results under identical simulation settings. Specifically, we train MiB with the same hyperparameters (e.g., batch size) as ours, except for the learning rate and the balancing parameters for the loss terms. We use exactly the same implementations for a loss function and the initialization technique as the ones in the official code of MiB. We present in Table R9 results for MiB under the (15-1) overlapped setting on PASCAL VOC [11]. All numbers are obtained by averaging results over five runs with standard deviations. We can see that our method clearly outperforms MiB in terms of $\text{mIoU}\_{\text{all}}$ and $\text{hIoU}$ scores. Note that the numbers in Table R9 for MiB are slightly lower than the numbers in Table R7. A plausible reason is that the hyperparameters (e.g., batch size) used in our simulation setting might not be optimal for other methods.
> > >
> > > \
> > > Table R9: Quantitative results on PASCAL VOC [11] for (15-1) overlapped settings.
> > > | | **$\text{mIoU}_{\text{b}}$**  | $\text{mIoU}_{\text{n}}$ | $\text{hIoU}$ |  $\text{mIoU}_{\text{all}}$ |
> > > |-|:-:|:-:|:-:|:-:|
> > > | $\text{MiB}$  | 31.68 $\pm$ 10.72 | 16.88 $\pm$ 1.86 | 21.66 $\pm$ 4.31 | 28.16 $\pm$ 8.50 |
> > > | $\text{Ours}$ | 78.09 $\pm$ 0.32 | 42.72 $\pm$ 1.58 | **55.21 $\pm$ 1.33** | **69.67 $\pm$ 0.49** |

---

> ### Author Response · Authors · 2022-08-02
> **Response to Reviewer X4eN [1/3]**
>
> We appreciate your detailed comments and suggestions. We will do our best to address your concerns. The major comments are answered below.
> ***
> **1. Hyperparameter analysis**
>
> Please see ‘Response to Reviewer vbQC: 2. Hyperparameter analysis’.
> ***
> **2. Applicability of our framework**
>
> We agree with your point that our initialization technique requires a training process for the auxiliary classifier before incremental steps and it is not applicable to off-the-shelf semantic segmentation models. We assume that current off-the-shelf segmentation models could be trained, as if novel object classes are likely to be added even after finishing the training process. That is, we could be aware that novel object classes will be added in the future, when training the models at the very first time. In this case, training the auxiliary classifier together is negligible. We will add the limitation to the limitation section. It is however worth noting that our framework without the initialization technique can be applied to off-the-shelf semantic segmentation models. Furthermore, our framework even without the initialization technique performs better compared to state-of-the-art methods [5, 9, 21] (See the results in Table 1 and the second row in Table 3). Note that our framework even without the initialization technique also outperforms recently proposed RCNet [C]. See the response to the reviewer vbQC (‘4. Comparison with RCNet’).
>
> We also agree with your point that “from an applicability perspective ours is rather comparable to replay-based approaches”, e.g., SSUL-M [6]. From Tables 1 and 2, ours without using a subset of training samples achieves comparable performance or even outperforms SSUL-M. Note that SSUL-M also exploits an off-the-shelf saliency detector [15]. Note also that the memory footprint of our method is much smaller than SSUL-M. More specifically, the memory footprint of weights for the auxiliary classifier is significantly smaller than a subset of images and ground-truth label maps. Similar to replay-based approaches, our method can exploit external memory that stores a subset of training samples. From Tables 1 and 2, ours using the external memory (Ours-M) outperforms SSUL-M on all class-incremental semantic segmentation benchmarks. Note that our framework also outperforms RECALL [A] for all class-incremental semantic segmentation benchmarks (See ‘4. Comparison with RECALL’ for details).
>
> Facilitating class-incremental learning (CIL) has many advantages for real-world applications, and our *incremental-aware learning* prior to incremental steps will provide a novel paradigm for CIL. We have achieved this by simply employing an auxiliary classifier for training. This requires negligible computational overhead, while achieving competitive performance among recent class-incremental semantic segmentation methods [5, 6, 9, 21, C].

---

### Official Review · Reviewer_vbQC · 2022-07-08

**Rating:** 4
**Confidence:** 4
**Soundness:** 3 good
**Presentation:** 3 good
**Contribution:** 2 fair

**Summary:**

This paper proposes a CISS framework which decomposes the prediction logit into two terms.  Then the DKD technique and an auxiliary classifier is adopted to improve the rigidity and transfer the knowledge of negatives, respectively. Experimental results demonstrate the effectiveness of the proposed framework.

**Questions:**

1. One contribution of this paper is to propose the decomposed knowledge distillation (DKD) loss, which distills the knowledge of the decomposition. And the decomposition values are obtained by decomposing the prediction logit into two scores (i.e., element-wise sum of positive or negative elements). The decomposition strategy is unconvincing, please provide the theoretical proof.
2. In Equation 1, the hyperparameters and are necessary to the entire model. How are they selected in the experiments? Please provide the experiments about the hyperparameters. And dynamic selection will be a good choice to balance the hyperparameters.
3. Both KD loss and DKD loss help the model retain the old knowledge, and is there any conflict between them? Please add the ablation study.
4. Please compare the proposed algorithm with the state-of-arts method [1].
5. Please add experiments on the cityscapes dataset [2].
6. The current manuscript need to be carefully polished.

[1] Chang-Bin Zhang, Jia-Wen Xiao, Xialei Liu, Ying-Cong Chen, Ming-Ming Cheng. Representation Compensation Networks for Continual Semantic Segmentation. In Proceedings of the IEEE Conference on Computer Vision and Pattern Recognition (CVPR), 2022.
[2] M. Cordts, M. Omran, S. Ramos, T. Reheld, M. Enzweiler, R. Benenson, U. Franke, S. Roth, and B. Schiele. The cityscapes dataset for semantic urban scene understanding. In Proceedings of the IEEE Conference on Computer Vision and Pattern Recognition (CVPR), 2016.

**Ethics Review Area:**

["I don’t know"]

**Limitations:**

+The first limitation is whether the framework can still perform well on a large number of task.
+The second limitation is that the hyperparameters in formula (1) are not dynamic.


**Strengths And Weaknesses:**

Strengths:
+ This paper proposes a new distillation method, DKD to address the forgetting problem.
+ Extensive comparison experiments illustrate the effective perofrmance of the proposed model to some degree.
+ The main idea of this paper is easy to understand.

Weaknesses:
+The contributions of this paper are somewhat limited, which cannot meet the requirements of NeurIPS acceptance.
+The motivation of some parts of this paper (distillation and initialization) needs to provide more clear explanations.
+It would be beeter to discuss how the backbone architecture affects continual learning performance.

---

> ### Author Response · Authors · 2022-08-02
> **Response to Reviewer vbQC [3/3]**
>
> **4. Comparison with RCNet**
>
> We thank you for suggesting an additional reference. We show in Tables R5 and R6 quantitative comparisons between ours and RCNet [A] on PASCAL VOC [11] and ADE20K [28], respectively. We can see that our method outperforms RCNet for all class-incremental semantic segmentation benchmarks. We will add the results of RCNet in Tables 1 and 2.
>
> \
> Table R5: Quantitative results on PASCAL VOC [11] for overlapped settings.
> | 15-5 (2 steps)| $\text{mIoU}_{\text{b}}$  | $\text{mIoU}_{\text{n}}$ | $\text{hIoU}$ |  $\text{mIoU}_{\text{all}}$ |
> |-|:-:|:-:|:-:|:-:|
> | $\text{RCNet}$| 78.80 | 52.00 | 62.65 | 72.40 |
> | $\text{Ours}$ | 78.83 $\pm$ 0.23 | 58.23 $\pm$ 0.45 | **66.98 $\pm$ 0.31** | **73.93 $\pm$ 0.21** |
>
> | **15-1 (6 steps)**| $\text{mIoU}_{\text{b}}$  | $\text{mIoU}_{\text{n}}$ | $\text{hIoU}$ |  $\text{mIoU}_{\text{all}}$ |
> |-|:-:|:-:|:-:|:-:|
> | $\text{RCNet}$| 70.60 | 23.70 | 35.49 | 59.4 |
> | $\text{Ours}$ | 78.09 $\pm$ 0.32 | 42.72 $\pm$ 1.58 | **55.21 $\pm$ 1.33** | **69.67 $\pm$ 0.49** |
>
> | 10-1 (11 steps)| $\text{mIoU}_{\text{b}}$  | $\text{mIoU}_{\text{n}}$ | $\text{hIoU}$ |  $\text{mIoU}_{\text{all}}$ |
> |-|:-:|:-:|:-:|:-:|
> | $\text{RCNet}$| 55.40 | 15.10 | 23.73 | 34.3 |
> | $\text{Ours}$ | 73.10 $\pm$ 0.56 | 46.51 $\pm$ 1.27 | **56.84 $\pm$ 0.97** | **60.44 $\pm$ 0.69** |
>
> \
> Table R6: Quantitative results on ADE20K [28] for overlapped settings.
> |100-50 (2 steps)| $\text{mIoU}_{\text{b}}$  | $\text{mIoU}_{\text{n}}$ | $\text{hIoU}$ |  $\text{mIoU}_{\text{all}}$ |
> |-|:-:|:-:|:-:|:-:|
> | $\text{RCNet}$| 42.30 | 18.80 | 26.03 | 34.50 |
> | $\text{Ours}$ | 42.41 $\pm$ 0.42 | 22.89 $\pm$ 0.37 | **29.74 $\pm$ 0.40** | **35.95 $\pm$ 0.38** |
>
> |100-10 (6 steps)| $\text{mIoU}_{\text{b}}$  | $\text{mIoU}_{\text{n}}$ | $\text{hIoU}$ |  $\text{mIoU}_{\text{all}}$ |
> |-|:-:|:-:|:-:|:-:|
> | $\text{RCNet}$| 39.30 | 17.60 | 24.31 | 32.10 |
> | $\text{Ours}$ | 41.56 $\pm$ 0.36 | 19.51 $\pm$ 0.35 | **26.55 $\pm$ 0.32** | **34.26 $\pm$ 0.24** |
>
> |50-50 (3 steps)| $\text{mIoU}_{\text{b}}$  | $\text{mIoU}_{\text{n}}$ | $\text{hIoU}$ |  $\text{mIoU}_{\text{all}}$ |
> |-|:-:|:-:|:-:|:-:|
> | $\text{RCNet}$| 48.30 | 25.00 | 32.95 | 32.50 |
> | $\text{Ours}$ | 48.84 $\pm$ 0.34 | 26.28 $\pm$ 0.60 | **34.17 $\pm$ 0.52** | **33.90 $\pm$ 0.43** |
> ***
> **5. Additional experiments on domain-incremental learning**
>
> The experiment provided in PLOP [9] using Cityscapes [B] is a benchmark for the *domain-incremental* learning scenario, which is beyond the scope of our work. Although we agree with your point that applying our method to domain-incremental learning could strengthen the contributions, we are currently focusing on *class-incremental* semantic segmentation (CISS), similar to other recent methods [5, 6, 21].
> ***
> **6. Performance on a large number of tasks**
>
> We have provided in Table S1 in the supplementary material a quantitative comparison between ours and state-of-the-art CISS methods [5, 6, 9] on PASCAL VOC [11]. All models are trained under an overlapped setting with the scenarios of (10-1) and (5-3). We can clearly see from Table S1 that our method outperforms the state-of-the-art methods in terms of $\text{mIoU}_{\text{all}}$ and $\text{hIoU}$ scores, confirming that our framework still performs well on a large number of tasks.
> ***
> References
> - [A] Representation Compensation Networks for Continual Semantic Segmentation, In CVPR, 2022.
> - [B] The cityscapes dataset for semantic urban scene understanding. In CVPR, 2016.

---

> ### Author Response · Authors · 2022-08-02
> **Response to Reviewer vbQC [2/3]**
>
> **2. Hyperparameter analysis**
>
> We fix $\beta$ to 0, and vary the value of $\alpha$ within {0,1,5,10}. We show in Table R3 results for different values of $\alpha$ under the (15-1) overlapped setting on PASCAL VOC [11]. All numbers are obtained by averaging results over five runs with standard deviations. From the first row in Table R3, we can see that training a CISS model without using both the KD and DKD terms in Eq. (1) causes catastrophic forgetting, drastically deteriorating the performance. We can see that employing KD is particularly effective in alleviating the forgetting problem. We have empirically set $\alpha$ to 5 for all experiments.
>
> \
> Table R3: Quantitative comparison for variants of the value of $\alpha$.
> | $\alpha$ | **$\text{mIoU}_{\text{b}}$**  | $\text{mIoU}_{\text{n}}$ | $\text{hIoU}$ |  $\text{mIoU}_{\text{all}}$ |
> |-|-:|:-:|:-:|:-:|
> | 0   | 21.31 $\pm$ 3.43 |   2.27 $\pm$ 1.88 | 4.11 $\pm$ 3.07 | 16.78 $\pm$ 2.86 |
> | 1   | 60.13 $\pm$ 4.75 | 28.28 $\pm$ 6.20 | 38.47 $\pm$ 6.68 | 52.55 $\pm$ 4.96 |
> | 5   | 74.43 $\pm$ 1.15 | 39.41 $\pm$ 1.51 | **51.53 $\pm$ 1.53** | **66.09 $\pm$ 1.19** |
> | 10 | 66.64 $\pm$ 8.10 | 36.36 $\pm$ 2.33 | 46.97 $\pm$ 3.85 | 59.40 $\pm$ 6.66 |
>
> \
> We also vary the value of $\beta$ between {0,1,5,10} while $\alpha$ is set to 5. We present in Table R4 results for different values of $\beta$ under the overlapped setting with the scenarios of (15-1) on PASCAL VOC. All numbers are obtained by averaging results over five runs with standard deviations. We can see that setting $\beta$ to be a positive value always provides better results. This validates the effectiveness of the DKD term. We can also see from Table R4 that our method is robust to various choices of $\beta$.
>
> \
> Table R4: Quantitative comparison for variants of the value of $\beta$.
> | $\beta$ | **$\text{mIoU}_{\text{b}}$**  | $\text{mIoU}_{\text{n}}$ | $\text{hIoU}$ |  $\text{mIoU}_{\text{all}}$ |
> |-|-:|:-:|:-:|:-:|
> | 0   | 74.43 $\pm$ 1.15 | 39.41 $\pm$ 1.51 | 51.53 $\pm$ 1.53 | 66.09 $\pm$ 1.19 |
> | 1   | 77.57 $\pm$ 0.57 | 42.14 $\pm$ 1.39 | 54.62 $\pm$ 1.20 | 69.14 $\pm$ 0.59 |
> | 5   | 78.09 $\pm$ 0.32 | 42.72 $\pm$ 1.58 | **55.23 $\pm$ 1.33** | **69.67 $\pm$ 0.49** |
> | 10 | 76.93 $\pm$ 0.85 | 41.30 $\pm$ 2.32 | 53.75 $\pm$ 2.08 | 68.45 $\pm$ 1.04 |
> ***
> **3. Conflict between KD and DKD**
>
> Both DKD and KD terms attempt to address catastrophic forgetting, and the two terms are complementary to each other. More specifically, we can see that KD alleviates the forgetting problem from Table R3. From Table R4, we can see that applying DKD further boosts the performance.

---

> ### Author Response · Authors · 2022-08-02
> **Response to Reviewer vbQC [1/3]**
>
> Thank you for taking the time to review our work. We will do our best to address your concerns. The major comments are answered below.
> ***
> **1. Additional explanation for DKD**
>
> Both KD and DKD techniques share the same objective, that is, keeping network weights similar to ones of an old model. This is because the network weights should not be updated drastically in learning novel classes in order to avoid catastrophic forgetting. Following your suggestion, we present a theoretical support for the necessity of the DKD technique. To this end, we first analyze the gradient of the KD loss w.r.t the output logit as follows:
>
> $\frac{\partial \mathcal{L}\_{\text{kd}}(i,c)}{\partial z_{t}(i,c)} = p_{t}(i,c) - p_{t-1}(i,c).$
>
> Let us suppose that
>
> $z_{t}^{+}(i,c) - z_{t-1}^{+}(i,c) = \delta$ and $z_{t}^{-}(i,c) - z_{t-1}^{-}(i,c) = -\epsilon$, and $\delta, \epsilon \neq 0$.
>
> Then,
>
> $z_{t}^{+}(i,c) + z_{t}^{-}(i,c)  - (z_{t-1}^{+}(i,c)+z_{t-1}^{-}(i,c)) = \delta - \epsilon$.
>
> That is,
>
> $z_{t}(i,c)  - z_{t-1}(i,c) = \delta - \epsilon$.
>
> The KD technique encourages the output logit $z_{t}(i,c)$ to be $z_{t-1}(i,c)$ obtained from an old model, making $\delta$ and $\epsilon$ to be equal. This also enforce the gradient w.r.t the output logit $\frac{\partial\mathcal{L}\_{\text{kd}}(i,c)}{\partial z\_{t}(i,c)}$ to be zero, suggesting that the output logit along with positive/negative reasoning scores is not affected by the KD loss. Thus, the actual difference between $z_{t-1}^{+}(i,c)$ and $z_{t}^{+}(i,c)$ ($z_{t-1}^{-}(i,c)$ and $z_{t}^{-}(i,c)$) still exists after applying the KD loss. Since the reasoning scores are crucial in quantifying how likely and unlikely an input belongs to a particular class, it is important to retain each reasoning score separately in order to preserve the discriminative ability for old classes.
>
> The DKD technique achieves this by explicitly imposing a constraint on each reasoning score. We also analyze the gradient of the DKD loss w.r.t each reasoning score as follows:
>
> $\frac{\partial \mathcal{L}\_{\text{dkd}}^{+}(i,c)} {\partial z_{t}^{+}(i,c)} = p_{t}^{+}(i,c) - p_{t-1}^{+}(i,c), \quad \frac{\partial\mathcal{L}\_{\text{dkd}}^{-}(i,c)}{\partial z_{t}^{-}(i,c)} = p_{t}^{-}(i,c) - p_{t-1}^{-}(i,c) $
>
> We can clearly see that the gradients w.r.t the reasoning scores, $z_{t}^{+}(i,c)$ and $z_{t}^{-}(i,c)$, are not zero, even when $\delta$ and $\epsilon$ are the same. Note that the DKD term encourages both $\delta$ and $\epsilon$ to be zeros, mitigating the catastrophic forgetting by keeping network weights similar to ones of an old model effectively. It is worth noting that we have also demonstrated empirically in Fig. 4 that KD does not preserve both positive and negative scores, while DKD prevents abrupt changes of the reasoning scores effectively (L290-292).

---

### Official Review · Reviewer_HGuy · 2022-07-11

**Rating:** 6
**Confidence:** 4
**Soundness:** 3 good
**Presentation:** 3 good
**Contribution:** 3 good

**Summary:**

The paper addresses incremental learning applied to semantic segmentation. There are two main contributions: adding an extra loss term aiming at preserving separately the positive and negative contribution values in the classification logit and learning an additional classifier for negative sample prediction that can be transferred to initialize the new classifier with incremented categories. Evaluation is done on two classical benchmarks: PASCAL VOC and ADE20k.


**Questions:**

I do not have any precise question, only a request for more justification of the two main contributions (see the weaknesses point).


**Limitations:**

Non applicable

**Strengths And Weaknesses:**

Strengths

- The paper is clearly written, and mostly well motivated (see however my remarks below).

- The ablation study, experiments and result analysis justify the approach on standard benchmarks.


Weaknesses

- Finding a good loss for class imbalance management is an old problem in ML (e.g., the focal loss that has been used in object detection or AUC maximization). The approach should have been compared, at least in principle, to solutions that have been proposed in the literature outside the field of incremental learning.

- The auxiliary classifier learned to initialize the next incremental step seems to be the most original idea of the paper, and should have been better developed and studied. The impact  of initialization, as illustrated in a few examples in Figure 5, seems to be strong and would have deserved a more in depth analysis, and, possibly, a more conceptual justification. Somehow, I am disappointed that the focus has been on the loss addressing class imbalance – there are other ways to handle it, and this is a well known phenomenon of incremental learning – and not on the initialization step.

---

> ### Author Response · Authors · 2022-08-02
> **Response to Reviewer HGuy**
>
> Thank you for your constructive comments. We will do our best to address your concerns. The major comments are answered below.
> ***
> **1. Relation between DKD technique and class imbalance problem**
>
> We would like to clarify that our approach does not necessarily address the class imbalance problem. Class imbalanced learning attempts to learn knowledge of rare classes, where corresponding training samples are insufficient compared to other ones. On the other hand, class incremental learning approaches focus on preserving knowledge of old classes, even when the corresponding supervisory signals are not available during training. The DKD technique has proven to be particularly effective to preserve knowledge of old classes for class-incremental semantic segmentation (CISS).
>
> Moreover, we would like to clarify that the class imbalance problem is problematic for memory-based CISS methods [2, 4, 6, 16, 26]  (L87-89), and mitigating the issue is not our main focus. Since our method facilitates learning without external memory that stores training samples explicitly, the detrimental effect of the imbalance problem is negligible compared to the catastrophic forgetting within our framework.
> ***
> **2. Additional analysis for initialization technique**
>
> We show in Table R1 additional experimental results for our method with and without the initialization technique under the (19-1) overlapped setting on PASCAL VOC [11]. All numbers are obtained by averaging results over five runs with standard deviations. From Table R1, we can see that applying the initialization technique gives a large $\text{mIoU}_{\text{n}}$ gain of 9.01%, confirming once more the effectiveness of the initialization technique.
>
> \
> Table R1: Quantitative comparison for variants of our method under the (19-1) overlapped setting on PASCAL VOC [11].
> | Baseline | Initialization | $\mathcal{L}\_{\text{dkd}}$ |\| | $\text{mIoU}_{\text{b}}$ | $\text{mIoU}_{\text{n}}$ | $\text{hIoU}$ |  $\text{mIoU}_{\text{all}}$ |
> |:-:|:-:|:-:|:-:|:-:|:-:|:-:|:-:|
> | V | Rand.| V | \| | 77.89 $\pm$ 0.10 | 32.44 $\pm$ 5.42 | 45.59 $\pm$ 5.58 | 75.73 $\pm$ 0.28 |
> | V | Ours| V | \| | 77.76 $\pm$ 0.18 | 41.45 $\pm$ 2.91 | **54.03 $\pm$ 2.49** | **76.03 $\pm$ 0.24** |
>
> \
> To further analyze the effectiveness of the initialization technique, we present in Table R2 recall and precision scores for a tv class, which belongs to a novel class in the (19-1) overlapped setting on PASCAL VOC. All numbers are also obtained by averaging results over five runs with standard deviations. Recall and precision are measured by $\frac{N_{TP}}{N_{TP} + N_{FN}}$ and $\frac{N_{TP}}{N_{TP} + N_{FP}}$, respectively, where $N_{TP}$, $N_{FP}$, and $N_{FN}$ are the number of true positives, false positives, and false negatives, respectively. We can see that our methods with and without the initialization technique show similar recall scores, suggesting that both classify tv objects well as a tv class, even without the initialization technique.
>
> On the other hand, our method with the initialization technique outperforms its counterpart in terms of the precision score. This indicates that the initialization technique reduces the number of false positives and explains the reason why the initialization technique boosts the $\text{mIoU}_{\text{n}}$ scores in Table R1. These results demonstrate once more the effectiveness of our initialization technique that allows a novel classifier to learn from abundant negative samples in previous learning steps, improving the discriminative ability of the novel classifier.
>
> \
> Table R2: Quantitative comparison for variants of our method under the (19-1) overlapped setting on PASCAL VOC [11].
> | Baseline | Initialization | $\mathcal{L}\_{\text{dkd}}$ | \| | Recall [%] | Precision [%] |
> |:-:|:-:|:-:|:-:|:-:|:-:|
> | V | Rand.| V | \| | 78.88 $\pm$ 0.93 | 35.60 $\pm$ 6.41 |
> | V | Ours| V | \| | 78.40 $\pm$ 1.06 | 46.80 $\pm$ 3.43 |

---

### Meta-Review · Area_Chair_5gJa · 2022-08-28

**Recommendation:** Accept
**Confidence:** Less certain

**Metareview:**

This submission deals with incremental semantic segmentation. The authors propose two contributions: 1/ a new knowledge distillation loss based on two separate loss functions for positive and negative class logits; 2/ a dedicated initialization strategy for the classifiers of novel classes. They present strong results on several standard benchmarks for incremental semantic segmentation.
This submission received initial diverging ratings. Reviewers have raised important concerns about the strategy and some missing experiments. After rebuttal, the active reviewers appreciated the answers and additional experiences provided. Following discussions, the final scores of active reviewers have increased and are clearly positive on this submission.

On the whole, the novelty and the interest of the proposal stand out clearly. The AC agrees that the strengths in this case outweigh the weaknesses. Authors are encouraged to consider all comments for their final version.

**Award:**

No

---

### Decision · Program_Chairs · 2022-09-14

Accept